# Fast and Stable Responses during Decision Making Require Strong Inhibitory Processes in Soccer Players

**DOI:** 10.3390/brainsci14030199

**Published:** 2024-02-22

**Authors:** Takahiro Matsutake, Hiroki Nakata, Genta Matsuo, Takayuki Natsuhara, Kisho Zippo, Kouki Watanabe, Takayuki Sugo

**Affiliations:** 1Research Center for Urban Health and Sports, Osaka Metropolitan University, Osaka 558-8585, Japan; 2Faculty of Engineering, Nara Women’s University, Nara 630-8263, Japan; hiroki-nakata@cc.nara-wu.ac.jp; 3Department of Sport Sciences, Osaka University of Health and Sport Sciences, Osaka 590-0459, Japan; gmatsuo@ouhs.ac.jp (G.M.); sugo@ouhs.ac.jp (T.S.); 4Faculty of Applied Psychology, Tokyo Seitoku University, Chiba 276-0013, Japan; t-natsuhara@tsu.ac.jp; 5Department of Sport Medicine and Research, Japan Institute of Sports Sciences, Japan High Performance Sport Center, Tokyo 115-0056, Japan; kisho.jippo@jpnsport.go.jp; 6Graduate School of Sport and Exercise Sciences, Osaka University of Health and Sport Sciences, Osaka 590-0459, Japan; 222m32@ouhs.ac.jp

**Keywords:** event-related potentials, contingent negative variations, N2, P3, Go/No-go, reaction time, soccer

## Abstract

Quick decision making is a vital factor for a successful pass in soccer games. Many previous studies of decision making in soccer focused on motor execution, but inhibitory processing has remained unclear. This study aimed to clarify the differences in motor execution and inhibitory among Japanese collegiate soccer players with different skill levels. We evaluated the behavioral data and event-related potentials in the high-skilled, low-skilled, and novice groups during the Go/No-go and pass choice reaction tasks. The reaction time (RT) was significantly shorter in the high group than in the novice group, and RT variability was small in the high group. The amplitude of the N2 component was significantly larger in the high group than in the low and novice groups, and the latency of the P3 component was significantly shorter in the high and low groups than in the novice group during the pass choice reaction task. The subtracted No-go N2 amplitude was also significantly larger in the high and low groups than in the novice group, and correlations existed between the RT, RT variability, and the subtracted No-go N2 amplitude during these tasks. These data indicate that soccer players’ behavioral responses and inhibition processing decision-making activities are associated with skill levels.

## 1. Introduction

Deciding whether to pass at a soccer game is complex, as various information and quick responses are vital factors for a successful pass. The percentage of high-intensity play is increasing in professional leagues [1], requiring players to perceive or cognize the location of a successful pass by extracting anticipatory cues from the positions of teammates and opponents. The difficulty and frequency of this action with the skill needed to perform at even a miss have led to the hypothesis that skilled soccer players, like other high-performing athletes, have specific perceptual and cognitive abilities relative to non-athletes. The potential neural correlation that indicates perceptual and cognitive performance enhancements specific to soccer players has been previously investigated. Notably, event-related potentials (ERPs) obtained using time-locked averaging electroencephalography (EEG) with high temporal resolution have been used. For example, Wang et al. [2] compared 30 male collegiate soccer players and 30 physically active male non-player novices using ERP and background alpha EEG to explore the neural mechanisms underlying perceptual decisions. A clear redundancy gain was observed on ERP/EEG data selectively for soccer players but not for control participants. The redundancy gain on the P3 component was positively associated with workload capacity over faster responses. These findings suggest that soccer players and control participants differ in processing multiple information sources in the presence of distractors. Iwadate et al. [3] investigated the effects of long-term training on the somatosensory ERPs between soccer players and non-athletes in upper- and lower-limb-targeted oddball tasks. They showed that P300 latency was significantly shorter in soccer players than in non-athletes during the lower limb task but not during the upper limb task. Notably, although research demonstrates that athletes outperform non-athletes in attention measures and cognitive processing speed [4,5,6,7,8,9], no conclusive evidence exists except for two previous studies [2,3] to support the processing speed advantage in soccer players. In addition, previous studies comparing soccer players and non-players did not examine any preparatory neural differences during the pre-stimulus interval. Notably, some studies have investigated the amount of motor preparation and other cognitive processes in baseball and table tennis players [4,5]. Hung et al. [4] found differences in the broadband power of table tennis players by measuring the ERP during their preparatory period (contingent negative variations (CNVs)) before a cued stimulus. Muraskin et al. [5] found that baseball players had significantly greater CNV amplitudes than novices, and higher CNV amplitudes correlated with better performance. A CNV is associated with motor preparation and cognitive processes, including attention, expectancy, motivation, and arousal [10,11,12]. This study analyzed the neural preparatory signals of CNVs to compare the pre-stimulus responses of experts and novices.

Soccer players require a quick response and need to stop the movement in the pass action as quickly as possible to decide whether to pass at a soccer game. This processing is called “inhibition”, which actively suppresses the motor command to execute actions. Therefore, we assume that skill-specific training improves inhibitory processes in skilled soccer players. The Go/No-go task helps to clarify the motor execution and inhibition processing. In addition, the neural mechanisms for the inhibitory process are investigated by recording ERPs while performing the Go/No-go task. In the No-go trials, the N2 and P3 components at approximately 140–300 ms and 300–500 ms are prominently elicited at the frontocentral electrodes compared with the Go trials [13,14,15,16]. N2 is generated from the anterior midcingulate and inferior frontal sources, and the generator constellation underlying P3 covers the precentral, middle frontal, and midcingulate areas [17]. Our previous study revealed that a stronger inhibitory cerebral activity is required for a No-go stimulus, whereas a stronger response is required for a Go stimulus. We also suggested that the neural activity for the inhibitory process is closely associated with muscle force, response speed, and variability [18]. For athletes, the amplitude of No-go N2 was larger in baseball players than in athletes in sports, and a significant positive correlation existed between the latency of No-go N2 and Go/No-go reaction time (RT) [19]. Similar findings for N2 amplitude were observed in top-level fencers [6]. These data suggest that athletes who require stimulus discrimination and motor response selection have faster responses facilitated by higher attention and stronger inhibitory activity. To our knowledge, motor execution and inhibition are vital for soccer playing; however, we do not know of any studies examining such neural characteristics in soccer players. In addition, we inferred that another likely difference between expert soccer players and novices was their respective abilities for task-specific perception–action coupling. Perception–action coupling involves tightly integrating perceptual processing with action generation [20]. This could be linked to neural substrate development that improves with training, enabling rapid and reliable predictions from incoming perceptual information. Therefore, measuring ERPs in a real environment or simulating the playing situation is vital for understanding soccer players’ motor execution and inhibition processing.

Based on these backgrounds, this study aimed to investigate the differences in motor execution and inhibition between skilled and nonskilled soccer players and novices in a pass choice behavior simulated task in soccer. Previous studies showed that the Go/No-go RT was significantly shorter in higher-skill baseball players than in lower-skill players [21]. The amplitudes of the P3 and pN2 components over central–parietal areas during tennis-specific tasks were larger in expert tennis players than in less expert tennis players [8]. Conversely, motor execution and inhibition activities could differ depending on skill level; therefore, participants in this study were classified into three groups based on their soccer skill level. Using the RT and ERPs, Nakamoto and Mori [22] also reported that skilled baseball players have conceptually linked the baseball batting-specific spatial information and response execution or inhibition. Therefore, for a detailed understanding of soccer players’ motor execution and inhibition processing, soccer-specific and general, non-soccer-specific tasks, are necessary. We examined RT and ERP components during the two Go/No-go tasks using different stimulus images (red or blue and whether to pass). We hypothesized that if skilled soccer players conceptually link soccer-specific spatial information and responses, a faster response would be observed with higher attention (including motor preparation) and stronger inhibition activity. Following a previous study, we also inferred the bivariate correlative relationship between behavioral data and ERPs, including the amplitudes of CNV and No-go-related components during soccer-specific tasks [19]. These data would advance the understanding of motor execution and inhibitory processing in soccer players.

## 2. Materials and Methods

### 2.1. Participants

Fourteen male amateur collegiate soccer players (20.7 ± 1.7 years) and seven male graduate students (22.3 ± 1.3 years) participated in this study. The collegiate soccer players were further divided into two groups based on their skill levels to investigate the differences between skilled and nonskilled soccer players. The high-skill group (high group) included seven collegiate soccer players (age 20.5 ± 1.5 years) who belonged to the official university team and were regular Japanese Kansai University Soccer League members. This team had previously won a Japanese university soccer championship. The low-skill group (low group) included seven collegiate soccer players (age 19.3 ± 0.5 years) who had never competed at the national level (in Japan) or participated in prefectural or regional competitions. Seven male graduate students with no experience in soccer training made up the novice group. Participants were recruited through the experimenters. None of the participants had any history of neurological or psychiatric disorders. The procedures used followed the Declaration of Helsinki regarding human experimentation. Written informed consent was obtained from all the participants before the experiment. This study was approved by the Ethical Committee of Osaka University of Health and Sports Science, Osaka, Japan.

### 2.2. Task and Procedure

The participants performed the visual Go/No-go and Go/No-go tasks, which simulated soccer decision making (pass choice reaction task). The two tasks were performed in the order of the Go/No-go task and the pass choice reaction task. Participants kept their eyes open throughout each task and focused on a fixation point on the monitor (1920 × 1080 px) positioned in front of them at approximately 1.1 m. Visual stimuli were presented using still image/video stimulation software (FMT-PV; Four Assist, Tokyo, Japan). After the advance notice stimulus (Stimuli-1: S-1) was presented on the screen for 2000 ms, an image of the Go or the No-go stimulus (Stimuli-2: S-2), an imperative stimulus, was presented for 1000 ms (Figure 1). In the Go/No-go task (Figure 1A), the Go and No-go stimuli were red and white circles presented at the center of the monitor with a black background. The probability of the Go and No-go stimuli was the same in a random series, with a 5000 ms inter-trial interval. Participants had to respond to the stimulus by pushing a foot switch button with their dominant foot as quickly as possible only after presenting the Go stimulus. Each session comprised 60 trials consisting of 30 No-go and 30 Go trials.

In the pass choice reaction task (Figure 1B), after S-1 was presented from the screen for 2000 ms, an image with five players (three offenders and two defenders) was presented for 1000 ms for the pass reaction task, which was S-2. In S-2, the Go stimulus was an image in which the ball was passed to the player in front of the two defenders. Participants had to respond to the stimulus by pushing a foot switch button with their dominant foot as quickly as possible only after presenting the Go stimulus. The No-go stimulus was an image in which the ball was passed to the players on the left or right. The simulated view included the first-person view of the ball holder in a 4 vs. 2 ball possession training at the just point of the decision making in pass choice. For those unfamiliar with soccer, the three images differed in the position of the defenders and receivers. During the 4 vs. 2 ball possession training, we aimed to keep the ball from being stolen by a defender and to pass the ball between the two defenders. Therefore, selecting and executing passes by observing the movements of defenders and receivers was necessary. Following previous research on soccer and preliminary investigations [23,24], the images were taken from a height of 2 m above the ground and 10 m away from where the soccer player played. Four licensed Japanese Football Association coaches who reviewed all the images taken ensured the validity of the pass selection images. Twelve approved images were used for the test (five and eight Go and No-go stimulation images, respectively).

### 2.3. EEG Recording

EEG was recorded using Ag/AgCl disk electrodes placed at the Fz, Cz, Pz, F3, F4, C3, C4, P3, and P4 points according to the 10–20 system. Each scalp electrode was referenced to the linked earlobes and calculated as an average reference. An electrooculogram was bi-polarly recorded to reject eye movements or blinks >100 μV, with a pair of electrodes placed 2 cm lateral to the lateral canthus of the left eye and 2 cm above the upper edge of the left orbit. In addition, we checked all raw data offline, and the trials were eliminated from averaging if the recorded clear artifacts did not exceed 100 μV. Impedance was maintained at <5 kΩ. Notably, all EEG signals were collected using a biosignal processor (MaP8100; NIHON SANTEKU Co., Ltd., Osaka, Japan). The recording epoch for ERPs was 200 ms before the S-1 onset and 1000 ms after the S-2 onset (3200 ms in total). The bandpass filter was set at 0.05–100 Hz, and the sampling rate was 1000 Hz. The baseline CNV period was measured from 200 ms before the S-1 to S-1 onset. CNV amplitudes were analyzed from 1500 to 2000 ms after the S-1 onset as late CNVs. The baseline period for the N2 and P3 components was calculated from 200 ms before the S-2 to S-2 onset because the N2 and P3 components were recorded after the S-2 stimuli presentation. The peak amplitudes and latencies of N2 and P3 were measured at 180–400 and 300–500 ms, respectively. Amplitudes were measured at the baseline to peak. Slow responses >800 ms and incorrect responses were eliminated from the averaging. In each task, at least 20 trials or more were averaged. Behavioral data, RT, the standard deviation (SD) of the RT (reaction time variability), omission (slow response or no pushing with the Go stimulus), and commission errors (pushing error with the No-go stimulus) were evaluated for each task.

### 2.4. Statistical Analysis

For behavioral data, RT, SD of RT, omission, and commission errors were compared using two-way repeated measure analysis of variances (ANOVAs) with the task (Go/No-go task vs. pass choice reaction task) as a within-subject factor and group (high, low, and novice) as a between-subject factor. For analyzing ERPs, amplitudes of the late CNV were submitted to a three-way repeated measure ANOVA with the task and electrode (Fz, Cz, and Pz) as within-subject factors and the group as a between-subject factor. Amplitudes and latencies of the N2 and P3 were analyzed using a four-way repeated measure ANOVA with the task, stimulus (Go vs. No-go), and electrode as with-in-subject factors and the group as a between-subject factor.

In addition, to extract the No-go-specific neural activities, we analyzed the difference waveform resulting from the subtraction of the ERP evoked by the Go stimulus from that evoked by the No-go stimulus, as the potentials that overlap between Go and No-go ERPs can be removed [8,19,25]. The subtracted waveform exhibited two prominent peaks, negative and positive peaks at 250–350 ms (subtracted No-go N2) and 350–510 ms (subtracted-No-go P3) after stimulus onset, respectively. Since each peak was clearly recorded at all participants’ frontal electrodes (Fz, F3, and F4), the data at these three electrodes were analyzed using a three-way repeated measure ANOVA with the task and electrode as within-subject factors and the group as a between-subject factor. If Mauchly’s test result was significant and the assumption of sphericity was violated, the Greenhouse–Geisser adjustment was used to correct for sphericity by altering the degrees of freedom using a correction coefficient (epsilon). The Bonferroni post hoc multiple comparison was performed to assess differences among groups and electrodes.

Furthermore, we analyzed the bivariate correlations (Pearson’s correlation) between the behavioral responses, amplitudes, and latencies of subtracted No-go N2 and subtracted No-go P3 at the frontal electrodes (Fz, F3, and F4). Statistical significance was set at *p* < 0.05. Statistical analyses were performed using SPSS (SPSS for Windows ver. 22.0).

## 3. Results

### 3.1. Behavioral Data

Table 1 shows the mean RT, SD of the RT, and commission errors. The results of the ANOVAs for the RT showed a significant main effect of the task (F (1, 18) = 109.983, *p* < 0.001, η^2^ = 0.859), indicating a shorter RT during the Go/No-go task than during the pass choice reaction task. ANOVAs for the RT also showed a significant main effect of the group (F (2, 18) = 2.989, *p* = 0.076, η^2^ = 0.249). A post hoc Bonferroni test showed that the RT was significantly shorter in the high group than in the novice group (*p* < 0.05). ANOVAs for the SD of the RT showed significant task–group interactions (F (2, 18) = 4.748, *p* = 0.020, η^2^ = 0.345). A post hoc Bonferroni test of this interaction showed that the SD of the RT during the pass choice reaction task was significantly larger in the low group than in the high group (*p* < 0.05). In contrast, the SD of the RT in the novice and low groups was significantly larger during the pass choice reaction task than during the Go/No-go task (*p* < 0.001). However, no significant differences were observed between tasks in the high group. No significant main effects or interactions were observed regarding omission and commission errors.

### 3.2. CNVs

Figure 2 shows the grand-averaged waveforms of CNVs during the Go/No-go and pass choice reaction tasks. The mean values of the late CNV amplitudes are listed in Table 2. The ANOVAs for the late CNV amplitude showed a significant task–electrode interaction (F (2, 18) = 6.533, *p* = 0.020, η^2^ = 0.278). The post hoc Bonferroni test of this interaction showed that the late CNV amplitudes were significantly larger at Cz than at Fz and Pz in both tasks (*p* < 0.001, respectively). No other significant effects or interactions were observed (Table 2).

### 3.3. N2 and P3 Components

Figure 3 shows the grand-averaged waveforms of the ERPs during the Go/No-go and pass choice reaction tasks. ERP components N2 and P3 were elicited during each task and were confirmed in 20 of 21 participants. The mean values of the N2 and P3 amplitudes and latencies are listed in Table 3.

The ANOVAs for the N2 amplitude showed significant task–stimulus (F (1, 17) = 5.368, *p* = 0.033, η^2^ = 0.240), stimulus–electrode (F (1.46, 24.89) = 7.526, *p* = 0.006, η^2^ = 0.307), and group–electrode interactions (F (4, 68) = 2.646, *p* = 0.050, η^2^ = 0.237). The post hoc Bonferroni test of task–stimulus interaction showed that the Go N2 and No-go N2 amplitudes were significantly larger during the pass choice reaction task than during the Go/No-go task (*p* < 0.001, respectively). The post hoc Bonferroni test of stimulus–electrode interaction showed that the Go N2 amplitudes were significantly larger at Cz than at Fz and Pz (*p* < 0.001, respectively), and No-go N2 amplitudes were significantly larger at Cz than at Pz (*p* < 0.05). The post hoc Bonferroni test of the group–electrode interaction showed that the N2 amplitudes were novice groups in Cz (*p* < 0.01).

Regarding N2 latency, no significant main effects or interactions were observed.

The ANOVAs for P3 amplitude showed significant task–stimulus–electrode interaction (F (2, 34) = 16.622, *p* < 0.001, η^2^ = 0.494) and stimulus–electrode interaction (F (2, 34) = 18.806, *p* < 0.001, η^2^ = 0.525). The post hoc Bonferroni test of stimulus–electrode interaction showed that the Go P3 amplitudes were significantly larger at Pz than at Fz and Cz (*p* < 0.001, respectively), and No-go P3 amplitudes were significantly larger at Cz than at Fz and Pz (*p* < 0.05, respectively). Additionally, the No-go P3 amplitudes were significantly larger than the Go P3 amplitudes at Fz and Cz (*p* < 0.05, respectively).

The ANOVAs for P3 latency showed significant task–group interaction (F (2, 17) = 10.603, *p* = 0.001, η^2^ = 0.555) and stimulus–electrode interaction (F (2, 34) = 3.454, *p* = 0.043, η^2^ = 0.169). The post hoc Bonferroni test of the task–group interaction showed that the P3 latencies were significantly shorter during the Go/No-go task than during the pass choice reaction task in all groups (*p* < 0.001, respectively), and P3 latencies were significantly shorter in the high and low groups than in the novice group in the pass choice reaction task (*p* < 0.001, respectively). The post hoc Bonferroni test of stimulus–electrode interaction showed that Go P3 latencies were significantly shorter than No-go P3 latencies at all electrodes (*p* < 0.01, respectively). 

### 3.4. Subtracted Waveform

Figure 4 shows the subtracted waveforms at all electrodes for each task and group. Large negative potentials after 250 ms and positive potentials after 350 ms were observed in the subtracted waveform, particularly for the frontal electrodes. The subtracted No-go N2 and subtracted No-go P3 components were confirmed in 19 of 21 participants (Fz, F3, and F4). Table 4 shows the mean peak latency and amplitude of subtracted No-go N2 and No-go P3.

The ANOVAs for the subtracted No-go N2 amplitude showed a significant main effect of task (F (1, 16) = 9.853, *p* = 0.006, η^2^ = 0.381), indicating a larger amplitude during the Go/No-go task than during the pass choice reaction task. The ANOVAs for the subtracted No-go N2 amplitude also showed a significant main effect of the group (F (1, 16) = 4.323, *p* = 0.032, η^2^ = 0.351). The post hoc Bonferroni test of the group showed that the subtracted No-go N2 amplitude was significantly larger in the high and low groups than in the novice group (*p* < 0.05, respectively). 

The ANOVAs for the subtracted No-go N2 latency showed a significant main effect of task (F (1, 16) = 27.124, *p* < 0.001, η^2^ = 0.629), indicating a shorter latency during the Go/No-go task than during the pass choice reaction task.

The ANOVAs for the subtracted No-go P3 amplitude showed a significant task–electrode interaction (F (2, 32) = 3.485, *p* = 0.043, η^2^ = 0.179). The post hoc Bonferroni test of this interaction showed that the subtracted No-go P3 amplitude was significantly larger during the Go/No-go task than during the pass choice reaction task at Fz and F4 (*p* < 0.01, respectively). The subtracted No-go P3 amplitude was significantly larger at Fz than at F3 and F4 in both tasks (*p* < 0.001, respectively).

The ANOVAs for the subtracted No-go P3 latency showed a significant main effect of the task (F (1, 16) = 17.042, *p* < 0.001, η^2^ = 0.516), indicating a shorter latency during the Go/No-go task than during the pass choice reaction task.

### 3.5. Relationship between RT, SD of RT, and Subtracted No-Go N2 and P3

The r values of the correlations between the RT, SD of RT, and the subtracted No-go components are listed in Table 5. During the pass choice reaction task, significant negative correlations were observed between the RT and the peak amplitude of the subtracted No-go N2 at Fz (r = −0.548; *p* = 0.012) and F3 (r = −0.549; *p* = 0.012), indicating that the peak amplitude of subtracted No-go N2 was larger for participants with a shorter RT (Figure 5A). A significant negative correlation was observed between the SD of the RT in the pass choice reaction task and the peak amplitude of the subtracted No-go N2 at Fz (r = −0.478, *p* = 0.033), indicating that the peak amplitude of subtracted No-go N2 was larger for participants with a small SD of the RT (Figure 5B). Except for these, no other correlations were observed between the RT, SD of RT, and the subtracted No-go N2 and P3.

## 4. Discussion

This study evaluated the differences in behavioral data (RT, SD of RT, and error rates) and ERP components (CNV, N2, and P3) among soccer players with different skill levels during the Go/No-go and pass choice reaction tasks. Behavioral data, including the RT, the SD of the RT, and the characteristics of the N2 and P3 components, were associated with skill levels. The subtracted No-go N2 amplitude was also significantly larger in the high and low groups than in the novice group. These data indicate that soccer players respond faster with the neural activity of stronger inhibition than novice players.

### 4.1. Behavioral Data

The RT was significantly shorter during the Go/No-go task than during the pass choice reaction task and was significantly shorter in the high group than in the novice group (Table 1). The RT is essential for understanding human sensorimotor performance [26]. It is defined as the time from stimulus onset to response, including components such as stimulus evaluation and response selection [27]. Previous studies on the RT in soccer players have reported that soccer players demonstrate faster performance than non-soccer players [28,29], consistent with the present study. Kosinski [30] also reported that a further complex task required additional stored information to elicit slower RT. Therefore, we assumed that the longer RT in our pass choice reaction task resulted from information elicitation and greater deliberation in the appropriate response for simulating the playing situation task. The SD of the RT in the novice and low groups was significantly larger during the pass choice reaction task than during the Go/No-go task. However, no significant effects were observed between tasks in the high group. The SD of the RT is often used to evaluate the reaction time variability between stimulus onset and response [31,32]. Previous studies have shown that participants who perform well on cognitive tasks tend to show less intra-individual variability and more stability than those who perform poorly [33,34]. Therefore, in the present study, although the pass choice reaction task influenced response variability in the novice and low groups, the high group might have higher stable responses.

### 4.2. CNVs

The CNV comprises two components: early frontocentral-dominant and late centroparietal-dominant components. The present study showed centroparietal distribution in the late CNV; no significant differences were observed among groups and between tasks (Table 2). As the introduction mentions, the CNV reflects motor preparation and cognitive processes, such as attention, expectancy, motivation, and arousal [10,11,12]. Previous studies on the CNV demonstrated the generator mechanisms in some brain regions, such as the prefrontal cortex, supplementary motor area, premotor area, and subcortical structures, including the basal ganglia and thalamus [35,36]. Our results suggest that the amount of motor preparation and other cognitive processes reflected by the CNV do not differ between skilled soccer players and novices.

### 4.3. N2 and P3 Components

Regarding N2 latency, no significant main effects or interactions were observed, whereas the N2 amplitude was larger in the pass choice reaction task than in the Go/No-go task. The N2 component is a member of a family of negative subcomponents, peaking at approximately 200–350 ms after stimulus onset. It is reliably enhanced for novel and further complex visual stimuli [37]. The difference in the N2 amplitude between the Go/No-go and pass choice reaction tasks would reflect complex visual stimuli in the pass choice reaction task. Previous studies have suggested that the No-go N2 originates from some regions, including the prefrontal cortex (PFC) and the anterior cingulate cortex (ACC) [18,38]. 

However, the site of the ACC regarding the source of the Go N2 is posterior to the source of the No-go N2 [6]. In addition, a previous study reported that baseball players and fencers had larger N2 amplitudes than non-athletes [6,19]. Consistent with this, the N2 amplitude in the present study was larger in the high group than in the low and novice groups with centroparietal distribution. This implies that soccer-specific training may modulate ERP components associated with higher-order brain functions, such as the response selection stage.

We showed that the No-go P3 amplitudes were larger than the Go P3 amplitudes, the parietal areas (Pz) were dominant in the Go P3, and the No-go P3 was central (Cz) dominant (Table 3). Previous studies using electrical dipole modeling and magnetoencephalography proved that P3 activity originates from multiple cerebral regions, such as the PFC, ACC, parietal cortex, temporalparietal junction, and hippocampal area [18,39]. The amplitude of No-go P3 was generally larger than that of Go P3, and No-go P3 shows a more anterior distribution relative to Go P3, the so-called “anteriorization” of No-go P3 [13,40]. Our previous study showed that the anteriorization of No-go P3 was weak in prepubescent children compared with adults, indicating that immature neural networks were related to No-go P3 [41]. In the present study, no significant main effect of group or interaction with a group factor was observed for the P3 amplitude; therefore, we assumed that the anteriorization of No-go P3 was not associated with the skill levels of soccer players.

P3 latency was significantly shorter during the Go/No-go task than during the pass choice reaction task (Table 3). Goodin et al. [42] reported that P3 latency increases when stimuli are difficult to discriminate or when the difficulty increases. This suggests that stimulus classification and evaluation were more difficult in the pass choice reaction task than in the Go/No-go task. In addition, the present study showed that the P3 latency was significantly shorter in the high and low groups than in the novice group during the pass choice reaction task. Studies on soccer players [2,3] have suggested that long-term training shortens the P300 latency in soccer players compared with that in non-athletes. Our data indicated no difference between soccer players and non-players in the Go/No-go task. In contrast, there was a difference in the stimulus evaluation speed in the pass choice reaction task. Consequently, whether this phenomenon is innate or learned is unclear; however, differences in the P3 latency between soccer players and non-players are present in soccer-specific perception–action.

### 4.4. Subtracted Waveform

Subtraction procedures are needed to study the potential effects of No-go [6,14,18,19]. Russo et al. [6] reported that fencers had a larger extracted No-go N2 amplitude by subtraction than controls in a visual Go/No-go task. Yamashiro et al. also [19] reported that a shorter RT was associated with a higher amplitude of No-go N2. Similarly, our study showed a larger amplitude of the subtracted No-go N2 in the high and low groups than in the novice group and a correlation between the RT (also SD of RT) and the amplitude of the No-go N2 in the pass choice reaction task. Therefore, soccer players appear to respond faster, with neural activity of stronger inhibition. These correlation results have an important implication. In the pass choice reaction task, a significant negative correlation was observed between the RT and the peak amplitude of the subtracted No-go N2; however, the Go/No-go task did not show any correlation. The participants were required to respond to red and white circles in the stimuli presented at the center in the Go/No-go task. 

However, they had to inhibit their response to at least two stimuli (left and right) based on the defender and receiver positions in the pass choice reaction task. This implies that soccer players could automatically activate their response to the inhibitory stimuli in the pass choice reaction task, which might have been caused by specific perception–action in soccer. Therefore, this suggests that the automatic activation of the response selection process caused RT facilitation in the high-skill group.

The amplitude of the subtraction No-go P3 did not differ between the groups. As mentioned above, the No-go N2 originates mainly from the PFC and ACC. However, the No-go P3 was associated with the neural activity of the PFC and ACC and the parietal cortex, temporalparietal junction, and hippocampal area [18,39]. A reasonable explanation for the differences between N2 and P3 may be the generator mechanisms.

### 4.5. Limitations

The limitations of this study included its small number of participants, and they were amateur players; therefore, in future studies, the number of subjects should be increased and the data of professional players should be examined to clarify the detailed differences in neural mechanisms of inhibition between skill levels.

## 5. Conclusions

Previous studies showed faster responses in soccer players during cognitive tasks than in non-athletes. However, this is the first study to elucidate faster and more stable responses with different soccer skill levels in situations where players need to decide whether to pass or not at a soccer game. We also showed specific neural activity of inhibition with highly skilled players. Our findings suggest that highly skilled soccer players are associated with the neural substrates of motor execution and inhibition processing.

## Figures and Tables

**Figure 1 brainsci-14-00199-f001:**
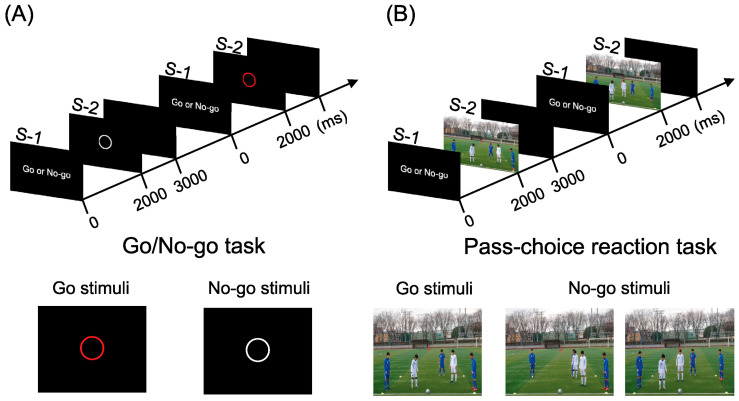
Scheme of experimental design in (**A**) the Go/No-go task and (**B**) the pass choice reaction task. In the Go/No-go task, the red circles represent the Go stimuli, and the white circles represent the No-go stimuli. In the pass choice reaction task, the Go stimuli is a situation where a pass can be made between two defense players, and the No-go stimuli is a situation where a pass can be given to the player on the left or right side.

**Figure 2 brainsci-14-00199-f002:**
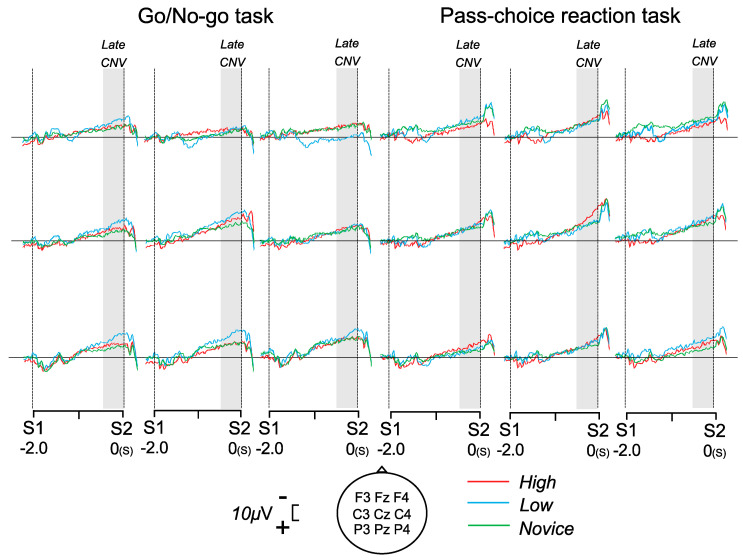
Grand-averaged waveforms of CNVs at nine electrodes for the high, low, and novice groups during the Go/No-go and pass choice reaction tasks.

**Figure 3 brainsci-14-00199-f003:**
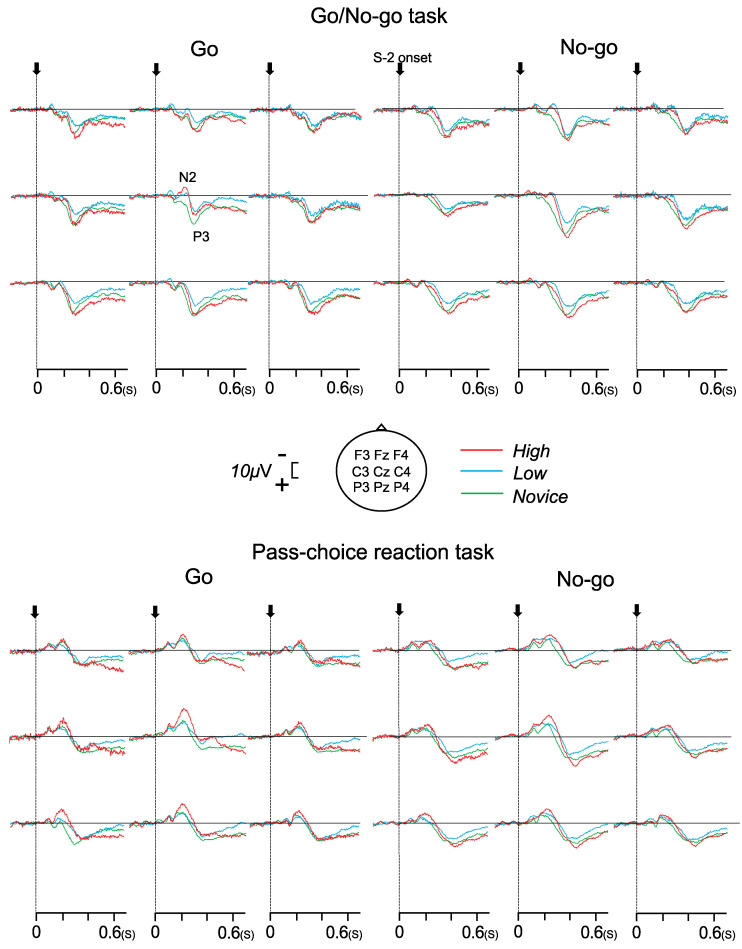
Grand-averaged ERPs after the S-2 presentation evoked by Go and No-go stimuli during the Go/No-go and pass choice reaction tasks.

**Figure 4 brainsci-14-00199-f004:**
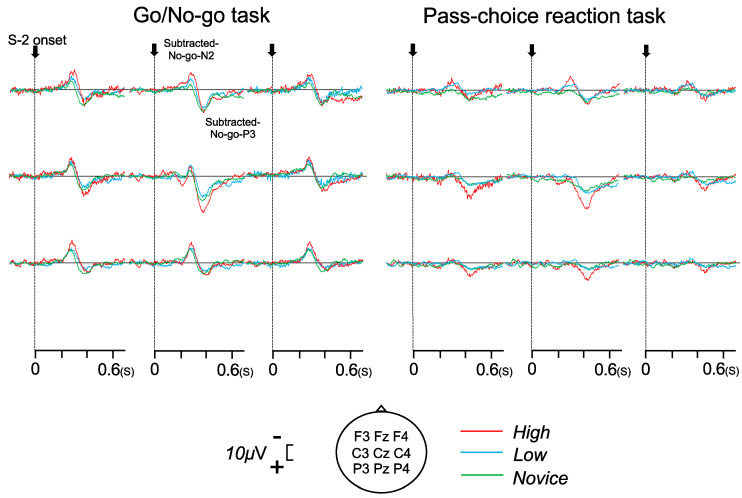
Subtracted waveform for the Go/No-go task (**left**) and pass choice reaction task (**right**).

**Figure 5 brainsci-14-00199-f005:**
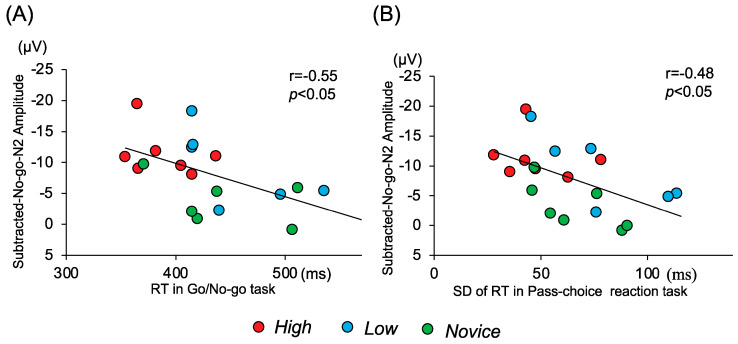
(**A**) Correlations between the RT and the subtracted No-go N2 amplitude at Fz during the Go/No-go task and (**B**) between the SD of the RT and the subtracted No-go N2 amplitude at Fz during the pass choice reaction task.

**Table 1 brainsci-14-00199-t001:** Behavioral data during Go/No-go and pass choice reaction tasks.

	Go/No-Go Task			Pass Choice Reaction Task		
	Novice	Low	High	Novice	Low	High
RT (ms)	348 ± 65	333 ± 30	311 ± 16 #	466 ± 78	447 ± 49	389 ± 30 #
SD of the RT (ms)	44 ± 21	47 ± 15	40 ± 11	66 ± 18 *	78 ± 25 *	48 ± 16 †
Omission errors (%)	0.0 ± 0.0	0.0 ± 0.0	0.0 ± 0.0	5.7 ± 0.1	0.0 ± 0.0	0.0 ± 0.0
Commission errors (%)	0.0 ± 0.0	0.0 ± 0.0	0.6 ± 1.0	1.8 ± 6.5	0.6 ± 1.0	0.0 ± 0.1

RT = reaction time; SD = standard deviation. Data are expressed as mean ± SD. Significant difference by ANOVAs; *: vs. Go/No-go task *p* < 0.05, †: vs. low group *p* < 0.01, #: vs. low group *p* < 0.05.

**Table 2 brainsci-14-00199-t002:** Mean amplitude for the late CNV at Fz, Cz, and Pz.

		Go/No-Go Task			Pass Choice Reaction Task		
Electrode		Novice	Low	High	Novice	Low	High
Late CNV (μV)	Fz	−3.0 ± 13.3	−5.8 ± 7.4	−6.1 ± 7.7	−13.2 ± 12.9	−7.9 ± 4.4	−7.7 ± 6.9
	Cz	−6.2 ± 15.8	−7.6 ± 6.5	−10.2 ± 6.8	−14.0 ± 12.6	−9.4 ± 5.3	−11.7 ± 7.6
	Pz	−4.6 ± 13.7	−6.0 ± 5.2	−6.1 ± 5.2	−10.2 ± 10.3	−5.7 ± 4.5	−6.7 ± 4.3

Data are expressed as mean ± SD.

**Table 3 brainsci-14-00199-t003:** Peak latencies and amplitudes of N2 and P3 at Fz, Cz, and Pz.

		**Go/No-Go Task**					
		**Go**			**No-Go**		
	**Electrode**	**Novice**	**Low**	**High**	**Novice**	**Low**	**High**
N2 Amplitude (μV)	Fz	0.5 ± 6.6	−1.7 ± 5.5	−1.1 ± 4.4	−0.3 ± 3.3	−6.3 ± 5.3	−4.2 ± 3.6
	Cz	0.3 ± 6.2	−2.9 ± 6.8	−9.3 ± 5.8	−0.4 ± 2.2	−3.9 ± 4.2	−5.6 ± 3.0
	Pz	0.5 ± 3.5	−1.2 ± 4.2	−3.2 ± 5.9	−1.6 ± 2.7	−1.7 ± 1.9	−3.4 ± 4.1
N2 Latency (ms)	Fz	237 ± 22	228 ± 36	209 ± 34	227 ± 31	235 ± 38	223 ± 31
	Cz	239 ± 31	220 ± 20	217 ± 25	229 ± 35	233 ± 37	212 ± 34
	Pz	238 ± 29	220 ± 21	205 ± 19	233 ± 28	223 ± 30	193 ± 16
P3 Amplitude (μV)	Fz	17.2 ± 6.1	14.1 ± 5.6	20.4 ± 7.2	23.5 ± 10.0	22.6 ± 9.3	23.2 ± 5.2
	Cz	23.9 ± 10.2	17.1 ± 5.0	20.8 ± 4.0	29.0 ± 10.8	23.3 ± 8.6	31.1 ± 4.5
	Pz	28.3 ± 10.0	21.9 ± 3.5	26.2 ± 6.4	24.4 ± 9.4	21.0 ± 7.5	26.2 ± 4.7
P3 Latency (ms)	Fz	332 ± 77	327 ± 30	340 ± 48	367 ± 19	374 ± 24	392 ± 39
	Cz	329 ± 79	342 ± 98	343 ± 57	360 ± 16	367 ± 20	379 ± 26
	Pz	328 ± 79	306 ± 8	326 ± 56	366 ± 16	379 ± 35	388 ± 37
		**Pass choice reaction task**					
		**Go**			**No-go**		
**Electrode**		**Novice**	**Low**	**High**	**Novice**	**Low**	**High**
N2 Amplitude (μV)	Fz	−11.9 ± 5.2	−12.6 ± 3.9	−13.9 ± 5.9	−10.3 ± 4.8	−11.9 ± 5.8	−13.5 ± 4.5
	Cz	−15.2 ± 4.9	−14.6 ± 3.2	−21.7 ± 4.8	−12.1 ± 5.3	−12.9 ± 5.2	−17.3 ± 5.2
	Pz	−11.7 ± 6.2	−11.5 ± 1.9	−15.2 ± 6.1	−8.9 ± 5.8	−10.5 ± 3.2	−12.8 ± 6.7
N2 Latency (ms)	Fz	237 ± 22	228 ± 36	209 ± 34	227 ± 31	235 ± 38	223 ± 31
	Cz	239 ± 31	220 ± 20	217 ± 25	229 ± 35	233 ± 37	212 ± 34
	Pz	238 ± 29	220 ± 21	205 ± 19	233 ± 28	223 ± 30	193 ± 16
P3 Amplitude (μV)	Fz	11.0 ± 7.8	8.7 ± 3.7	11.5 ± 4.2	14.9 ± 11.5	10.4 ± 8.4	13.5 ± 3.7
	Cz	12.3 ± 7.8	6.9 ± 4.3	9.6 ± 8.5	19.7 ± 11.2	13.6 ± 3.1	21.7 ± 7.4
	Pz	16.5 ± 7.3	11.2 ± 5.9	13.1 ± 10.2	18.1 ± 6.7	11.8 ± 3.1	18.6 ± 9.5
P3 Latency (ms)	Fz	467 ± 89	377 ± 22	376 ± 37	452 ± 47	426 ± 34	430 ± 21
	Cz	484 ± 62	378 ± 21	370 ± 40	446 ± 44	423 ± 35	431 ± 20
	Pz	476 ± 57	381 ± 35	370 ± 45	452 ± 46	451 ± 19	431 ± 33

Data are expressed as mean ± SD.

**Table 4 brainsci-14-00199-t004:** Peak latencies and amplitudes of subtracted No-go N2 and P3.

		Go/No-Go Task			Pass Choice Reaction Task		
Electrode		Novice	Low	High	Novice	Low	High
**Subtracted No-go N2 Amplitude (μV)**	**Fz**	−6.5 ± 5.3	−11.8 ± 5.3	−12.3 ± 6.9	−3.8 ± 3.8	−9.3 ± 6.1	−11.4 ± 3.7
	**F3**	−9.2 ± 4.6	−14.0 ± 4.2	−13.9 ± 5.3	−3.9 ± 3.1	−9.3 ± 5.8	−11.1 ± 5.0
	**F4**	−9.4 ± 4.9	−13.3 ± 1.9	−13.3 ± 4.7	−5.1 ± 3.8	−8.2 ± 3.5	−9.5 ± 2.5
**Subtracted No-go N2 Latency (ms)**	**Fz**	287 ± 17	285 ± 23	282 ± 53	343 ± 31	336 ± 24	320 ± 19
	**F3**	286 ± 17	287 ± 25	285 ± 50	339 ± 27	341 ± 25	322 ± 20
	**F4**	281 ± 38	284 ± 20	290 ± 66	347 ± 36	336 ± 25	319 ± 38
**Subtracted No-go P3 Amplitude (μV)**	**Fz**	20.4 ± 9.2	15.5 ± 9.3	17.2 ± 10.1	11.0 ± 8.0	11.2 ± 8.9	12.3 ± 6.1
	**F3**	16.4 ± 7.0	9.7 ± 10.4	13.4 ± 8.7	10.6 ± 8.0	7.8 ± 7.2	10.1 ± 4.5
	**F4**	15.3 ± 7.3	12.7 ± 8.0	14.5 ± 9.5	6.6 ± 5.3	9.8 ± 6.8	11.4 ± 5.5
**Subtracted No-go P3 Latency (ms)**	**Fz**	425 ± 90	388 ± 19	397 ± 18	448 ± 33	464 ± 47	445 ± 24
	**F3**	421 ± 90	392 ± 23	388 ± 16	506 ± 69	453 ± 34	430 ± 30
	**F4**	430 ± 85	407 ± 42	404 ± 16	415 ± 119	462 ± 39	425 ± 104

The data are expressed as mean ± SD.

**Table 5 brainsci-14-00199-t005:** The r values of correlations between RT, SD of RT, and subtracted No-go N2 and P3 at the three frontal electrodes.

			F3	Fz	F4
RT (Go/No-go task)					
Amplitude					
Subtracted No-go N2			−0.32	−0.22	−0.39
Subtracted No-go P3			0.20	0.22	0.36
Latency					
Subtracted No-go N2			0.09	0.15	−0.07
Subtracted No-go P3			−0.03	−0.05	−0.05
SD of the RT (Go/No-go task)					
Amplitude					
Subtracted No-go N2			−0.11	−0.24	−0.36
Subtracted No-go P3			0.06	−0.04	0.19
Latency					
Subtracted No-go N2			0.15	0.25	0.10
Subtracted No-go P3			−0.01	−0.10	−0.02
RT (pass choice reaction task)					
Amplitude					
Subtracted No-go N2			−0.55 *	−0.55 *	−0.34
Subtracted No-go P3			−0.09	0.09	0.25
Latency					
Subtracted No-go N2			0.36	0.20	0.29
Subtracted No-go P3			0.17	0.20	0.24
SD of the RT (pass choice reaction task)					
Amplitude					
Subtracted No-go N2			−0.40	−0.48 *	−0.39
Subtracted No-go P3			−0.26	−0.18	−0.06
Latency					
Subtracted No-go N2			0.22	0.10	0.18
Subtracted No-go P3			0.24	0.33	0.32

* *p* < 0.05.

## Data Availability

The data presented in this study are available on request from the corresponding author. The data are not publicly available due to privacy reasons.

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
