# Peer review of "Fast and Stable Responses during Decision Making Require Strong Inhibitory Processes in Soccer Players"

_brainsci, 2024, doi:10.3390/brainsci14030199_

Round 1
Reviewer 1 Report
Comments and Suggestions for Authors
The aim of this study was to clarify the differences in behavioral responses and brain processes among soccer players of various ability levels. The authors reported the differences in behavioral data and ERP components across the three groups of participants during the Go/No-go and pass-choice reaction tasks. It is a well-written and presented study that is certainly relevant to the special issue's topics. However, I would propose publishing it as pilot research. The electrophysiological and behavioral data are important although they were recorded by just 21 individuals divided into three groups. In my opinion, the small group of participants should be included in the limitations section. A minor recommendation is to correct some typos, such as in line 52 distraction and line 54 lower.
Author Response
Response to Reviewer #1
Thank you for this comment. We revised manuscript, as suggested by two reviewers, and we believe that the manuscript is proper improved.
However, I would propose publishing it as pilot research. The electrophysiological and behavioral data are important although they were recorded by just 21 individuals divided into three groups. In my opinion, the small group of participants should be included in the limitations section.
Thank you for this constructive comment. We revised limitations section (page 16, lines 561-564):
“The limitations of this study included small number of participants, and they were amateur players; therefore, in future studies, increase the number of subjects and the data of professional players should be examined to clarify the detailed differences in neural mechanisms of inhibition between skill levels.”
A minor recommendation is to correct some typos, such as in line 52 distraction and line 54 lower.
We are sorry for our mistakes. We corrected typos (page 2, lines 52,54).
Reviewer 2 Report
Comments and Suggestions for Authors
The topic of study seems very interesting to me,
I think that the authors should indicate in the abstract the context of the study, the country, the region and the city of the study.
In the method section the authors indicate
"El grupo de baja capacidad (grupo bajo) incluyó a siete jugadores de fútbol americano universitario (de 19.3 ± 0 años.que nunca habían competido a nivel nacional (en Japón) ni habían participado en competiciones prefecturales o regionales
They should indicate in more detail the profile of the participants and why they choose those players
En la sección de método, los autores en
The analyzes carried out are good.
The presentation of results is good.
Indicate that I ask the authors to see the following work as it will give them inspiration and will help them complete the introduction and, if possible, the discussions.
Martínez-González, J.M., Gil-Madrona, P., Carrillo-López, P.J., & Martínez-López, M. (2022). Perfectionism and affect as determinants of self-perceived motor competence in primary school children. Kinesiology, 54(2),
288-298. https://doi.org/10.26582/k.54.2.10
Author Response
Response to Reviewer #2
Thank you for this comment. We revised manuscript, as suggested by two reviewers, and we believe that the manuscript is proper improved.
I think that the authors should indicate in the abstract the context of the study, the country, the region and the city of the study.
We added the context of the study and country in the abstract (page 1, lines 17-20):
“Many previous studies of decision making in soccer focused on the motor execution, but inhibitory processing have remained unclear. This study aimed to clarify the differences in motor execution and inhibitory among Japanese collegiate soccer players with different skill levels.”
They should indicate in more detail the profile of the participants and why they choose those players.
We added the explanation of grouping (page 3, lines 124-133):.
“The collegiate soccer players were further divided into two groups based on their skill levels to investigate the differences between skilled and nonskilled soccer players. The high-skill group (high group) included seven collegiate soccer players (age 20.5±1.5 years) who belonged to the official university team and were regular Japanese Kansai University Soccer League members. This team had previously won a Japanese university soccer championship. The low-skill group (low group) included seven collegiate soccer players (age 19.3±0.5 years) who had never competed at the national level (in Japan) or participated in prefectural or regional competitions. Seven male graduate students with no experience in soccer training made up the novice group. Participants were recruited through the experimenters.”
Indicate that I ask the authors to see the following work as it will give them inspiration and will help them complete the introduction and, if possible, the discussions.
Thank you for recommendation. We checked this reference, as suggested. Unfortunately, however, this study mainly focused on the self-perceived motor competence, that the content and concept were different from our study. Therefore, we could not include this reference in the introduction and discussion.